# Adversarially Robust Learning for Security-Constrained Optimal Power Flow

**Priya L. Donti**\*, **Aayushya Agarwal**\*, **Neeraj Vijay Bedmutha, Larry Pileggi, J. Zico Kolter**
Carnegie Mellon University, Pittsburgh, PA, USA
{pdonti, aayushya, nbedmuth, pileggi, zkolter}@andrew.cmu.edu

## Abstract

In recent years, the ML community has seen surges of interest in both adversarially robust learning and implicit layers, but connections between these two areas have seldom been explored. In this work, we combine innovations from these areas to tackle the problem of N-k security-constrained optimal power flow (SCOPF). N-k SCOPF is a core problem for the operation of electrical grids, and aims to schedule power generation in a manner that is robust to potentially $k$ simultaneous equipment outages. Inspired by methods in adversarially robust training, we frame N-k SCOPF as a minimax optimization problem – viewing power generation settings as adjustable parameters and equipment outages as (adversarial) attacks – and solve this problem via gradient-based techniques. The loss function of this minimax problem involves resolving implicit equations representing grid physics and operational decisions, which we differentiate through via the implicit function theorem. We demonstrate the efficacy of our framework in solving N-3 SCOPF, which has traditionally been considered as prohibitively expensive to solve given that the problem size depends combinatorially on the number of potential outages.

## 1  Introduction

Robust optimization problems are pervasive across many applications and domains – such as electric power systems, supply chain management, and civil engineering – where the goal is to construct some solution that is robust under any allowable instantiation of uncertainty [1, 2]. While the aim is generally that these solutions be *provably* robust, there unfortunately remain many settings where it is either not easy or not possible to construct such solutions. This has often motivated the use of heuristic approaches. For instance, many approaches in adversarially robust deep learning formulate neural network training as a minimax game over neural network parameters and input perturbations, optimizing this problem via gradient-based techniques that do not yield provable robustness guarantees, but are nonetheless effective in practice [3].

In this work, we draw inspiration from adversarially robust training to address the problem of N-k security-constrained optimal power flow (SCOPF). N-k SCOPF is a fundamental problem to schedule power generation in a way that is robust to $k$ potential equipment failures (e.g., generator or line outages). Unfortunately, N-k SCOPF is prohibitively expensive to solve at scale, leading grid operators to use rough approximations in practice. To address this challenge, we frame N-k SCOPF as a minimax attacker-defender problem, where the "defender" aims to schedule power generation, and the "attacker" aims to pick adversarial equipment failures. The loss function of this problem requires solving implicit equations representing the physics of the electric grid as well as additional operational decisions that are made after an attack has occurred. As such, we optimize this problem using gradient-based techniques, and employ insights from the literature on implicit differentiation and implicit layers to cheaply compute gradients through the loss function.

---

\*These authors contributed equally.

35th Conference on Neural Information Processing Systems (NeurIPS 2021).

Our key contributions are:

- **Formulation for minimax optimization with implicit variables.** To streamline the presentation of concepts, we provide a generic formulation for gradient-based optimization of minimax problems with implicitly-defined variables. While our main focus in this paper is on N-k SCOPF, we believe this generic formulation may also be of broader interest for minimax settings with physics in the loop, as well as for tri-level optimization settings.

- **Formulation for gradient-based optimization of N-k SCOPF.** We rewrite N-k SCOPF as a continuous minimax optimization problem, and demonstrate how to efficiently compute gradients through relevant implicit components. We also utilize the underlying structure of our optimization solvers to further streamline the outer minimization procedure. Importantly, the per-iteration cost of this approach is agnostic to the number of simultaneous outages $k$, despite the combinatorial blowup in the number of associated "contingency scenarios."

- **Demonstration on N-1, N-2, and N-3 SCOPF.** We demonstrate the efficacy of our method in addressing SCOPF settings that allow for one, two, or three simultaneous outages on a realistic 4622-node power system with over 38 billion potential N-3 outage scenarios. We find that our method incurs 3-4$\times$ fewer N-3 feasibility violations than a baseline optimal power flow approach, and requires only 21 minutes to run on a standard laptop.

## 2   Related work

**Adversarial robustness in deep learning.** There has been a growing body of work that aims to parameterize neural networks in a manner that is robust to particular perturbations of their inputs, usually by casting neural network training as an attacker-defender game [3, 4]. While there have been several promising approaches for *certifiably* robust neural network training [5–7], in general, these approaches do not yet scale to large-sized networks and only address a limited set of threat models. As a result, there has been a lot of research in this area that aims to train robust neural networks using approximate, gradient-based training methods [8, 9], an approach we adopt in the context of SCOPF. In addition, a key part of this literature has been on constructing strong but cheap-to-compute attacks that can strengthen the outcomes of adversarially robust training, e.g., the fast gradient sign method (FGSM) [8] and projected gradient descent (PGD) attacks [9]. In our experiments, we similarly show how a gradient-based adversarial robustness approach can be used to identify potential grid vulnerabilities, as an input to secure power system optimization.

**Implicit layers.** Implicit differentiation techniques [10, 11] have started to be used more widely within deep learning workflows, largely in the context of *implicit layers*, i.e., neural network layers that represent implicit functions [12]. These include differentiable optimization layers [13–19] and physics-based layers [20, 21], among others [22, 23]. Given the large number of parameters within any given deep network, a key aspect of this work has been in finding efficient ways to actually compute the relevant derivatives, e.g., by strategically ordering multiplicative operations and reusing the results of previous computations. We similarly employ these kinds of strategies when computing implicit derivatives for our SCOPF optimization approach. We also note that some of the work on implicit differentiation in deep learning and related areas has been in service of solving bi-level optimization problems, e.g., for decision-driven forecasting [24, 25], hyperparameter tuning [26–29], or system identification and control [30]. We similarly consider the use of implicit differentiation for multi-level optimization (in particular, tri-level optimization) in the context of SCOPF.

**Security-constrained optimal power flow.** In the electric power systems community, there has been a great deal of emphasis on optimizing power grid operations to be secure to sets of outages (*contingencies*) that may be particularly high risk. For instance, many grid operators in the United States require grids to be operated in a way that is N-1 secure (i.e., secure against any single outage), which has led to a focus in the literature on addressing N-1 SCOPF [31–34]. However, ensuring security against *multiple* simultaneous failures (i.e., solving N-k SCOPF for $k > 1$) is becoming increasingly critical, both as evidenced by recent major blackout events [35, 36] and as climate change drives weather extremes [37] that may lead to correlated outages [38]. That said, due to the computational complexity of addressing N-k SCOPF in the general case, there have been few attempts at developing methods geared towards this setting. In particular, the computational complexity of N-k SCOPF grows combinatorially with $k$ and the size of the system. Some previous attempts to solve N-k SCOPF have employed exhaustive methods [39], Bender's cuts to reduce the number of

contingencies analyzed [40], and bi-level optimization frameworks [40, 41]. In particular, [41] used bi-level optimization to develop a systematic attacker-defender approach to address N-3 contingency scenarios, but used a simplified, linear power grid model to attain convergence. We similarly adopt a bi-level framework, but solve a realistic non-linear model of the grid by introducing fast gradient calculation methods inspired by the implicit layers literature, which allows us to scale our approach to a 4622-node system.

## 3   Generic problem formulation

Before diving into the details of our SCOPF formulation, we first provide a more generic formulation for gradient-based minimax optimization over an implicit loss function, which we will later build upon in the context of SCOPF. In particular, we consider the setting of continuous minimax optimization problems over "defender" (minimizer) variables $x \in \mathcal{X}$ and "attacker" (maximizer) variables $y \in \mathcal{Y}$; these are also referred to as first-stage and second-stage decision variables, respectively, in the bi-level optimization literature. In addition, we allow for "third-stage" decisions $z \in \mathcal{Z}$ that are fully defined via a set of implicit constraints on $x$, $y$, and $z$.

Specifically, we consider problems of the form

$$\underset{x \in \mathcal{X}}{\text{minimize}} \ \max_{y \in \mathcal{Y}} \ \ell(x, y, z)$$
$$\text{s.t.} \ \ g(x, y, z) = 0, \ \ z \in \mathcal{Z}, \tag{1}$$

where $\mathcal{X}$, $\mathcal{Y}$, and $\mathcal{Z}$ are compact sets; $\ell : \mathcal{X} \times \mathcal{Y} \times \mathcal{Z} \to \mathbb{R}$ is a standard, continuously differentiable loss function (e.g., softmax or mean squared error loss); and $g : \mathcal{X} \times \mathcal{Y} \times \mathcal{Z} \to \mathbb{R}^m$ is defined such that $g(x, y, z) = 0$ is an implicit function in $z$ with some solution $z \in \mathcal{Z}$ for all $(x, y) \in \mathcal{X} \times \mathcal{Y}$. We further restrict ourselves to those functions $g$ that are continuously differentiable with non-singular Jacobians at their roots, i.e., those functions that are compatible with the implicit function theorem [3, 42]. We note that this formulation covers a wide range of settings, e.g., many minimax problems with non-linear physical constraints, or many tri-level optimization problems where $z$ is a solution to a continuous optimization problem parameterized by $x$ and $y$ (both of which notions we will use in Section 4 for the setting of N-k SCOPF).

Inspired by the literature on adversarial robustness in deep learning, we propose to solve problem (1) via gradient-based search on both the inner maximization and outer minimization problems. In particular, this entails (a) obtaining some (approximately) optimal $y$ for the inner maximization problem via gradient-based techniques, given some initial value of $x$, (b) updating $x$ using the gradient at the optimum of the inner maximization problem, and (c) repeating these steps until convergence. We now describe steps (a) and (b) in additional detail.

### 3.1   Solving the inner maximization problem

Let $\bar{x}$ denote some fixed value for $x$. The inner maximization problem is then given by

$$\max_{y \in \mathcal{Y}} \ \ell(\bar{x}, y, z) \ \ \text{s.t.} \ \ g(\bar{x}, y, z) = 0, \ \ z \in \mathcal{Z}. \tag{2}$$

We optimize this problem via projected gradient descent. Specifically, let $y = y_0$ denote our initial guess for the optimal attack, and let $\mathcal{P}$ denote the projection operator. Until convergence (or for some fixed number of iterations), we then

(i)   Obtain $z^\star$ such that $g(\bar{x}, y, z^\star) = 0, \ z^\star \in \mathcal{Z}$.

(ii)   Update $y \leftarrow \mathcal{P}_{\mathcal{Y}} \left( y + \gamma \nabla_y \ell(\bar{x}, y, z^\star) \right)$ for step size $\gamma$.

Notably, step (ii) entails obtaining the gradient $\nabla_y \ell(\bar{x}, y, z^\star)$. By the chain rule, this involves the gradient through $z^\star$, which is the solution to a set of implicit equations. Specifically, using the notation $\mathrm{d}$ to denote total derivatives (e.g., gradients) and $\partial$ to denote partial derivatives, we have

$$\frac{\mathrm{d}\ell(\bar{x}, y, z^\star)}{\mathrm{d}y} = \frac{\partial\ell(\bar{x}, y, z^\star)}{\partial y} + \frac{\partial\ell(\bar{x}, y, z^\star)}{\partial z^\star} \frac{\mathrm{d}z^\star}{\mathrm{d}y}. \tag{3}$$

By the implicit function theorem, we can then obtain an expression for $\mathrm{d}z^\star/\mathrm{d}y$ by noting that

$$\frac{\mathrm{d}g(\bar{x}, y, z^\star)}{\mathrm{d}y} = \frac{\partial g(\bar{x}, y, z^\star)}{\partial y} + \frac{\partial g(\bar{x}, y, z^\star)}{\partial z^\star}\frac{\mathrm{d}z^\star}{\mathrm{d}y} = 0 \implies \frac{\mathrm{d}z^\star}{\mathrm{d}y} = -\left(\frac{\partial g(\bar{x}, y, z^\star)}{\partial z^\star}\right)^{-1}\frac{\partial g(\bar{x}, y, z^\star)}{\partial y}, \tag{4}$$

which we can plug into Equation (3) to yield our full update.

We note that in practice, we seldom want to compute the Jacobian $\mathrm{d}z^\star/\mathrm{d}y \in \mathbb{R}^{\dim(\mathcal{Z}) \times \dim(\mathcal{Y})}$ explicitly due to the potentially large time and space complexity of doing so; instead, it is often desirable to compute the left vector-matrix product $(\partial\ell/\partial z^\star)(\mathrm{d}z^\star/\mathrm{d}y) \in \mathbb{R}^{\dim(\mathcal{Y})}$ directly. We refer to this strategy as the "vector-Jacobian product trick." The details of the relevant computations may vary based on the particular setting at hand, and we describe how we employ this trick for SCOPF in Section 4.3.

### 3.2 Taking a gradient step in the minimization problem

Given some (approximately) optimal $y^\star$ and associated $z^\star$ from the inner optimization under the current value of $x = \bar{x}$, the outer optimization problem then becomes

$$\min_{x \in \mathcal{X}} \quad \ell(x, y^\star, z^\star) \quad \text{s.\,t.} \quad g(x, y^\star, z^\star) = 0. \tag{5}$$

One option is to then update $x$ via a projected gradient step $x \leftarrow \mathcal{P}_\mathcal{X}\left(x - \beta\nabla_x\ell(x, y^\star, z^\star)\right)$ for step size $\beta$. To calculate the gradient $\nabla_x\ell(x, y^\star, z^\star)$, we note that by Danskin's theorem, we can disregard the dependence of $y^\star$ on $x$ [3] (though we cannot ignore the dependence of $z^\star$). As such, we can employ a similar process as in Equations (3) and (4), where we treat $y^\star$ as constant when computing gradients with respect to $x$.[2] We note that while this is one potential process for updating $x$, we actually employ a more efficient, domain-specific process for our SCOPF procedure (see Section 4.4).

## 4 Addressing N-k SCOPF via adversarially robust optimization

Having presented this generic formulation, we now introduce our approach, CAN$\partial$Y, for addressing SCOPF.[3] In particular, we consider the problem of N-k SCOPF, where power generation must be scheduled so as to be feasible and low-cost both in the absence of equipment outages ("base case") as well as to be robust to any $k$ simultaneous outages of power generators or lines that may occur ("contingency cases"). We note that the set of *contingencies* – i.e., allowable combinations of outages – is combinatorial in the number of potential outages, making the SCOPF problem extremely computationally expensive. For instance, a realistic 4622 node system with 6133 potential single outages has ~38.5 billion contingency scenarios to consider under the N-3 setting.

In the rest of this section, we first more formally define the N-k SCOPF problem. We then show how we rewrite N-k SCOPF as a minimax problem of the form (1), in particular by forming a compact outer approximation to the contingency space. Finally, we describe how we solve this problem using a combination of gradient-based techniques and domain-specific enhancements, as summarized in Algorithm 1.

### 4.1 Defining N-k SCOPF

Let $x$ denote the *dispatch* – i.e., setpoints of *real power*[4] and voltage magnitude – at all power generators on the electricity system, and let $\mathcal{X}$ represent generator-wise box constraints on the dispatch. Let $\mathcal{C}$ denote the set of potential contingencies, i.e., all sets of exactly $k$ potential outages.

---

[2]Technically, Danskin's theorem only holds when $y^\star$ is a unique optimum of the inner maximization problem. However, in the adversarially robust training literature, the conditions of Danskin's theorem do not necessarily hold – in particular, the inner maximization problem often does not have a unique optimum, and many implementations tend to generate approximate (rather than exact) optima [8, 9] – but this method of computing gradients is used in practice regardless [3].

[3]CAN$\partial$Y stands for "CMU Adversarial Networks with Differentiable contingencY." This name is inspired by that of SUGAR [43], whose power flow solver we differentiate through in this work.

[4]Modern electric power systems are generally alternating current (AC) systems, in which all electrical quantities – e.g., powers and voltages – are considered to be complex-valued. In particular, the terms *real power* and *reactive power* refer to the real and imaginary components, respectively, of *complex power*.

---

**Algorithm 1** CAN∂Y

---
1: **procedure** MAIN(sys)   *// input: power system description*
2:     **init** dispatch $x$       *// e.g., via base case optimal power flow*
3:     **while** not converged **do**
4:         $y^\star$ = ATTACK(sys, $x$)   *// worst-case attack for current dispatch*
5:         **update** $x$ via partial solve of Equation (12) using Gauss-Siedel
6:     **end while**
7: **end procedure**
8:
9: **procedure** ATTACK(sys, $x$)
10:     **init** attack $y$
11:     **while** not converged *or* for fixed number of steps **do**
12:         **compute** $z^\star$, $s^\star$ via Equation (8c)          *// third-stage variables*
13:         **compute** $\nabla_y \ell(x, y, z^\star, s^\star)$ via Equation (11)   *// gradient of attack objective*
14:         **update** $y \leftarrow \mathcal{P}_\mathcal{Y} \left( y + \gamma \nabla_y \ell(x, y, z^\star, s^\star) \right)$   *// for attack set $\mathcal{Y}$, step size $\gamma$*
15:     **end while**
16:     **return** $y$
17: **end procedure**

---

Finally, let $z^{(i)} \in \mathcal{Z}_i(x, c^{(i)})$ represent slightly adjusted settings of real power and voltage magnitude that the power system operator can create after scheduling $x$ and then observing some contingency $c^{(i)} \in \mathcal{C}$, where the $\mathcal{Z}_i$ represent box constraints. Then, the N-k SCOPF problem can be expressed as

$$\underset{x \in \mathcal{X}}{\text{minimize}} \quad f_{\text{base}}(x) + \sum_{(z^{(i)}, c^{(i)})} f_{\text{cont}}(z^{(i)}, c^{(i)})$$

$$\text{subject to} \quad g_{\text{flow,base}}(x, w_{\text{base}}) = 0, \quad w_{\text{base}} \in \mathcal{W}_{\text{base}} \tag{6}$$

$$z^{(i)} \in \quad \begin{matrix} \text{argmin}_{z^{(i)} \in \mathcal{Z}_i(x, c^{(i)})} f_{\text{cont}}(z^{(i)}, c^{(i)}) \\ \text{s.\,t.}\ \ g_{\text{flow,cont}}(z^{(i)}, w^{(i)}, x) = 0, \ w^{(i)} \in \mathcal{W}_i(x, c^{(i)}) \end{matrix} \quad \forall c^{(i)} \in \mathcal{C},$$

where $f_{\text{base}} : \mathcal{X} \to \mathbb{R}$ represents base case power production costs; $g_{\text{flow,base}} : \mathcal{X} \times \mathcal{W}_{\text{base}} \to \mathbb{R}^{n_{\text{bus}}}$ represents the non-linear power flow equations in the base case, with $n_{\text{bus}}$ being the number of power system nodes; $w_{\text{base}}$ represents electrical quantities that result from solving the base case power flow equations (e.g., *reactive powers* and voltage angles), with box constraints (device limits) represented by $\mathcal{W}_{\text{base}}$; and $f_{\text{cont}} : \mathcal{Z}_i \times \mathcal{C} \to \mathbb{R}$, $g_{\text{flow,cont}} : \mathcal{Z}_i \times \mathcal{W}_i \times \mathcal{X} \to \mathbb{R}^n$, and $w^{(i)} \in \mathcal{W}_i(x, c^{(i)})$ represent their respective contingency-case counterparts. (See Appendix A for a more explicit formulation.)

## 4.2   Rewriting N-k SCOPF as a minimax problem

We reformulate the SCOPF problem (6) as an attacker-defender game, where the defender must choose a dispatch that is robust to potential "worst-case" contingencies chosen by an attacker. In particular, since the contingency set $\mathcal{C}$ is discrete, we create a continuous outer approximation to this set in order to enable the use of gradient-based techniques. Specifically, let $n_o$ be the number of generators or power lines that can potentially experience an outage. Then, for any $y \in [0,1]^{n_o}$, we define the $j$th entry as follows:

$$y_j = \begin{cases} 1 & \text{iff outage } j \text{ is fully active,} \\ 0 & \text{iff outage } j \text{ is not active,} \\ \alpha_j \in (0,1) & \text{iff outage } j \text{ is partially active with fraction } \alpha_j. \end{cases} \tag{7}$$

The first two notions presented in Equation (7) are standard in power systems: the generator or line pertinent to outage $j$ is either fully operational or out of service. We newly define the notion of a partial outage with fraction $\alpha_j$ as one in which the power flowing through the outage device during normal operation has been reduced by a factor of $\alpha_j$. For instance, we model a partial contingency on a transmission line or transformer device as reducing its admittance (i.e., ability to conduct current) by a factor of $\alpha_j$. Similarly, we restrict the power produced by a generator undergoing a partial contingency by multiplying its power output by $\alpha_j$.

Given these notions, we define our "threat model" for the N-k SCOPF setting to contain all vectors $y$ with an L1-norm of at most $k$, i.e., $\mathcal{Y} := \{y : y \in [0,1]^{n_o}, \|y\|_1 \leq k\}$. Notably, the original contingency set $\mathcal{C}$ is fully represented within $\mathcal{Y}$, and in fact, all scenarios with *up to $k$* simultaneous potential outages are also represented. As such, $\mathcal{Y}$ represents a much broader set of potential contingencies than specified in the original problem. (Relevantly for projected gradient descent, this is also a convex set.) Using this set, we can then write our reformulation of the SCOPF problem as

$$\underset{x \in \mathcal{X}}{\text{minimize}} \max_{y \in \mathcal{Y}} \quad f_{\text{base}}(x) + f_{\text{cont}}(z,y) + \frac{1}{2}\|s\|_2^2 \tag{8a}$$

$$\text{subject to} \quad g_{\text{flow,base}}(x, w_{\text{base}}) = 0, \quad w_{\text{base}} \in \mathcal{W}_{\text{base}} \tag{8b}$$

$$z, s \in \quad \begin{matrix} \text{argmin}_{z \in \mathcal{Z}(x,y),\, s \in \mathbb{R}^{n_{\text{bus}}}} \, f_{\text{cont}}(z,y) + \frac{1}{2}\|s\|_2^2 \\ \text{s.t. } g_{\text{flow,cont}}(z, w_{\text{cont}}, x) + s = 0, \; w_{\text{cont}} \in \mathcal{W}_{\text{cont}}(x,y), \end{matrix} \tag{8c}$$

where $s \in \mathcal{S} := \mathbb{R}^{2n_{\text{bus}}}$ are slack variables representing potential infeasibilities in the third-stage optimization problem, as necessitated by the expanded contingency set. In particular, the goal of the attacker is to now to find a set of partial outages that not only increase the cost of power generation, but also create instabilities in the grid, as captured by $s$. As we hinted at in Section 3, this is a minimax optimization problem with implicit constraints over the third-stage variables $z$, $w_{\text{base}}$, $w_{\text{cont}}$, and $s$, incorporating both non-linear equality constraints (8b) as well as an optimization-based constraint (8c).

### 4.3 Obtaining attack gradients

As described in Section 3.1, we aim to find the worst-case attack via projected gradient descent. In particular, we must compute the gradient of the minimax loss with respect to $y$, which is given by

$$\frac{\mathrm{d}\ell}{\mathrm{d}y} = \frac{\partial f_{\text{cont}}(z^\star, y)}{\partial y} + \frac{\partial f_{\text{cont}}(z^\star, y)}{\partial z^\star}\frac{\mathrm{d}z^\star}{\mathrm{d}y} + \frac{\mathrm{d}s^\star}{\mathrm{d}y}. \tag{9}$$

(As the base case power production cost and the base case power flow constraint (8b) have no dependence on $y$, we do not need to consider these terms during the inner maximization.)

To calculate the terms $\mathrm{d}z^\star/\mathrm{d}y$ and $\mathrm{d}s^\star/\mathrm{d}y$, we implicitly differentiate through the third-stage optimization problem (8c). In order to do so inexpensively, we reuse the results of computations that were executed when originally obtaining $z^\star$ and $s^\star$. More specifically, in order to obtain $z^\star$ and $s^\star$, we solve the non-linear KKT conditions of optimization problem (8c) using a Newton solver (see Appendix B), which entails linearizing these equations at each iteration. At convergence, we then implicitly differentiate through the linear fixed-point equation obtained at the last iteration:

$$J\begin{pmatrix} z^\star \\ s^\star \end{pmatrix} = b \implies \frac{\mathrm{d}J}{\mathrm{d}y}\begin{pmatrix} z^\star \\ s^\star \end{pmatrix} + J\begin{pmatrix} \frac{\mathrm{d}z^\star}{\mathrm{d}y} \\ \frac{\mathrm{d}s^\star}{\mathrm{d}y} \end{pmatrix} = \frac{\mathrm{d}b}{\mathrm{d}y} \implies \begin{pmatrix} \frac{\mathrm{d}z^\star}{\mathrm{d}y} \\ \frac{\mathrm{d}s^\star}{\mathrm{d}y} \end{pmatrix} = J^{-1}\left(-\frac{\mathrm{d}J}{\mathrm{d}y}\begin{pmatrix} z^\star \\ s^\star \end{pmatrix} + \frac{\mathrm{d}b}{\mathrm{d}y}\right), \tag{10}$$

where $J \in \mathbb{R}^{d \times d}$ is the Jacobian of the non-linear KKT system and $b \in \mathbb{R}^d$ is the corresponding right hand side vector, for $d = \dim(\mathcal{Z}) + \dim(\mathcal{S})$. We note that since the system Jacobian $J$ is in practice extremely sparse, the inverse term $J^{-1}$ can be computed extremely efficiently using sparse LU factorization; similarly, the higher-order derivative $\mathrm{d}J/\mathrm{d}y$ is extremely sparse and can be computed efficiently. We refer the reader to [44] for more details.

Substituting this result into Equation (9), the overall gradient of the loss is then given by

$$\frac{\mathrm{d}\ell}{\mathrm{d}y} = \frac{\partial f_{\text{cont}}(z^\star, y)}{\partial y} + \begin{pmatrix} \frac{\partial f_{\text{cont}}(z^\star,y)}{\partial z^\star} \\ 1 \end{pmatrix}^T J^{-1}\left(-\frac{\mathrm{d}J}{\mathrm{d}y}\begin{pmatrix} z^\star \\ s^\star \end{pmatrix} + \frac{\mathrm{d}b}{\mathrm{d}y}\right). \tag{11}$$

As hinted earlier, we employ the "vector-Jacobian product trick" in order to efficiently compute these gradients (see, e.g., [11]). In particular, rather than computing the terms $\mathrm{d}z^\star/\mathrm{d}y$ and $\mathrm{d}s^\star/\mathrm{d}y$ explicitly via Equation (10), we directly compute their left vector-matrix product with the relevant partial derivatives of the loss – i.e., the blue term in Equation (11), with multiplications evaluated from left to right to ensure we are always taking matrix-vector (rather than matrix-matrix) products. Inspired by [13], we also reuse the LU factor from the last Newton solve we computed when obtaining $z^\star$

and $s^\star$ in order to avoid explicitly (re-)computing the matrix inverse $J^{-1}$. Finally, we note that for this specific problem, while the last term $-\frac{\mathrm{d}J}{\mathrm{d}y}\begin{pmatrix} z^\star \\ s^\star \end{pmatrix} + \frac{\mathrm{d}b}{\mathrm{d}y}$ nominally involves tensor products, the relevant terms are vastly sparse due to the structure of the underlying physics – with no more than 20 nonzero entries per potential outage – making these products relatively cheap to compute in practice.

### 4.4 Solving the defense minimization

After obtaining a worst-case attack $y^\star$, our next step is to adjust our dispatch $x \in \mathcal{X}$ in response. As described in Section 3.2 for the generic setting, one option to do this involves taking a projected gradient step in $x$. However, we adopt a different approach for N-k SCOPF, due to the practical requirements of this setting. In particular, a general system requirement is that the base case power flow equations $g_{\text{flow,base}}(\cdot) = 0$ must remain feasible under the dispatch $x \in \mathcal{X}$, as the most likely scenario is that no contingency will occur. However, projecting onto this (non-linear, non-convex) set of constraints can be expensive.

As a result, we instead note that we can rewrite the N-k SCOPF minimax problem (8) as a single minimization problem:

$$
\begin{aligned}
\underset{x \in \mathcal{X},\, z \in \mathcal{Z}(x,y^\star),\, s \in \mathbb{R}^{n_{\text{bus}}}}{\text{minimize}} \quad & f_{\text{base}}(x) + f_{\text{cont}}(z, y^\star) + \frac{1}{2}\|s\|_2^2 \\
\text{subject to} \quad & g_{\text{flow,base}}(x, w_{\text{base}}) = 0, \quad w_{\text{base}} \in \mathcal{W}_{\text{base}} \\
& g_{\text{flow,cont}}(z, w_{\text{cont}}, x) + s = 0, \quad w_{\text{cont}} \in \mathcal{W}_{\text{cont}}(x, y^\star).
\end{aligned}
\tag{12}
$$

To determine our next iterate of $x$, our strategy is then to *partially* solve this optimization problem by running one step of a non-linear Gauss-Seidel method, and then keep the value of $x$ obtained from that step. This allows us to incrementally update $x$ in a direction that is more robust to the worst-case attack $y^\star$, while still maintaining the feasibility of the base case power flow equations.

Importantly, we are able to run this procedure efficiently, as we can reuse the results of existing computations that were executed when obtaining the optimal attack $y^\star$. In particular, as we describe in more detail in Appendix C, we can split the KKT conditions of the problem (12) into two groups:

$$
\begin{pmatrix} \mathrm{d}\mathcal{L}/\mathrm{d}x \\ \mathrm{d}\mathcal{L}/\mathrm{d}\lambda_{\text{base}} \end{pmatrix} \equiv F_{\text{base}}(x, w_{\text{base}}, \lambda_{\text{base}}) + \begin{pmatrix} \left(\partial g_{\text{flow,cont}}(z, w_{\text{cont}}, x)/\partial x\right)^T \lambda_{\text{cont}} \\ 0 \end{pmatrix} = 0
\tag{13a}
$$

$$
\begin{pmatrix} \mathrm{d}\mathcal{L}/\mathrm{d}z \\ \mathrm{d}\mathcal{L}/\mathrm{d}s \\ \mathrm{d}\mathcal{L}/\mathrm{d}\lambda_{\text{cont}} \end{pmatrix} \equiv F_{\text{cont}}(z, w_{\text{cont}}, \lambda_{\text{cont}}) + \begin{pmatrix} 0 \\ 0 \\ g_{\text{flow,cont}}(z, w_{\text{cont}}, x) \end{pmatrix} = 0,
\tag{13b}
$$

where $\mathcal{L}$ denotes the Lagrangian of problem (12), and $\lambda_{\text{base}}$ and $\lambda_{\text{cont}}$ are the dual variables on the base case and contingency power flow constraints, respectively. We note that the two terms $F_{\text{base}}$ and $F_{\text{cont}}$ are independent in terms of their inputs, but are weakly coupled via the additional terms (which represent a sparse set of ramping constraints and voltage setpoints that tie together the base and contingency cases). This is an ideal setup for decoupling through non-linear Gauss-Seidel solution methods. In addition, the contingency-related KKT conditions (13b) are actually identical to the KKT conditions of the problem (8c) that we solved during the last iteration of the inner maximization problem; as a result, we can reuse the result of this previous computation when executing our Gauss-Seidel step. Together, this allows us to inexpensively identify an update direction for $x$ that nonetheless remains feasible with respect to the base case power flow constraints.

## 5  Experiments

We demonstrate the efficacy of our approach on the settings of N-1, N-2 and N-3 SCOPF. In particular, noting that quality of adversarial attacks is likely to have a large effect on the success of our overall procedure, we first visualize the attacks found by our inner maximization process (Sections 3.1, 4.4) on a small power system test case. We then demonstrate the performance of our overall approach on a realistic 4622-node power system with approximately 6 thousand potential N-1 contingencies, almost 19 million N-2 contingencies, and over 38 *billion* N-3 contingencies. We show that CAN$\partial$Y

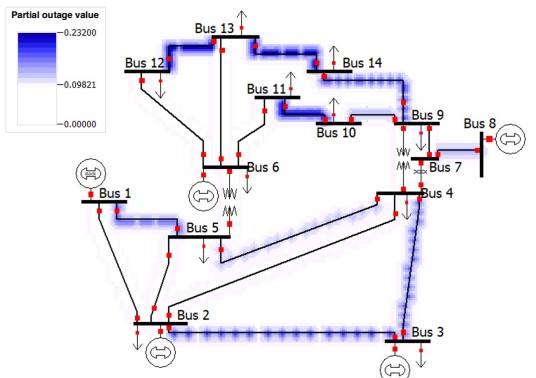

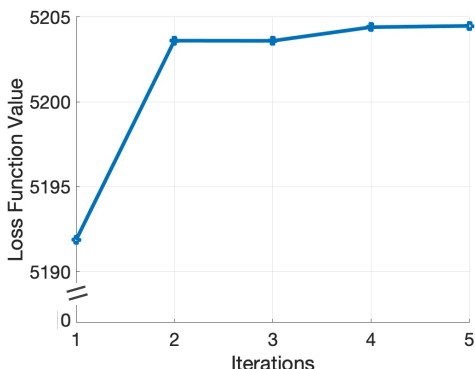

(a) Visualization of the worst-case contingency found, with the degree of the associated partial outage on each line indicated in blue. (Plot generated via PowerWorld.)

(b) Training curve for finding a worst-case contingency. The process converges within 5 iterations, and increases the loss by 3%.

Figure 1: Illustrative example of finding a worst-case N-2 contingency on a 14-node test system.

is able to efficiently find solutions that are competitive with other leading approaches in the N-1 case, while reducing violations in the N-2 and N-3 scenarios compared to a base case optimal power flow.

All experiments are run on a single core of a Macbook Pro with a 2.6 GHz Core i7 CPU. We implement our approach in Python, using a custom optimal power flow solver called SUGAR [45] to compute optimization (8c), and CVXPY [46] to compute convex projections for projected gradient descent. We evaluate all dispatch solutions using PowerWorld, a commercial power flow tool.

## 5.1 Illustrative adversarial attack

Finding worst-case contingencies is often beneficial to power systems engineers, who try to identify fragile areas of their grid for future development. Traditionally, engineers use linear approximations of the grid physics [47–49] to identify single outages that pose a significant risk to system stability. However, given that the underlying physics are fundamentally non-linear, such linear approximations quickly become inaccurate when trying to identify the risks associated with multiple simultaneous outages. Our implicit differentiation approach, on the other hand, employs accurate gradient information from the physics of the network to quickly identify contingencies that maximally increase our loss function (or an alternative loss function of choice that also captures system infeasibilities).

For ease of visualization, we demonstrate this attack-identification approach on the IEEE 14-node test system. In particular, we identify an adversarial N-2 contingency on this system in just 5 iterations (approximately one minute), increasing the value of the loss function by 3% over the base case scenario, as shown in Figure 1b. This worst-case contingency represents a combination of multiple partial outages on different lines, and (perhaps surprisingly) does not include any generator outages, as shown in Figure 1a. This is likely to present a stronger attack than those obtained via the "standard" linear approximation approach, serving as a potential benefit to power system planners who are trying to reinforce their grid, as well as to "adversarially robust training" procedures like ours.

## 5.2 Validating N-1 security

Today, most grid operators in the United States require that their dispatch be N-1 secure, i.e., secure against any single outage, prompting the development of associated methods. In particular, the recent Grid Optimization (GO) Competition [34], hosted by ARPA-E, focused on finding algorithms to solve N-1 SCOPF. Each participating team used a variety of methods to produce a dispatch that was evaluated on the basis of power cost and feasibility in both the base and contingency cases. In order to validate that our method works well in the N-1 setting, we solve a particular case from the competition – namely, the 4622-node test case with a sub-selection of 3071 N-1 potential contingencies, provided as part of the Challenge 1 stage – by constructing our relaxed contingency set $\mathcal{Y}$ with $k = 1$.

|  | gollnl | GO-SNIP | GMI-GO | BAT | gravityx | CAN∂Y* |
|---|---|---|---|---|---|---|
| GO Challenge 1 Rank | 1 | 2 | 3 | 4 | 5 | - |
| Score for 5K network | 546,302 | 553,152 | 553,328 | 545,783 | 550,020 | 552,032 |

Table 1: Comparison of the performance of our method against top-performing submissions to the ARPA-E GO Competition, which addresses N-1 SCOPF (lower scores are better). Results are shown for the 4622-node Challenge 1 test case ("5K network"). While the score comparisons shown are inexact due to subtleties of the evaluation metric (see Appendix D), at a high level, we see that CAN∂Y performs competitively with all top-scoring methods.

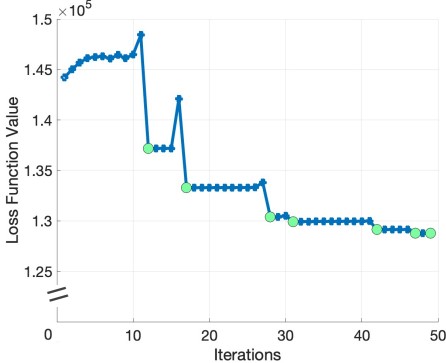

Figure 2: Loss vs. iterations in adversarial training. Each attack stage is at most 10 iterations and each defense stage is one iteration. The loss value after the defense step is in green.

| Contingency type | N-1 | N-2 | N-3 |
|---|---|---|---|
| Scenarios tested | 6,133 | 359,712 | 428,730 |
| OPF viol. | 59 | 10,572 | 4,086 |
| PowerModels viol. | 37 | 4,005 | 5,391 |
| CAN∂Y viol. (ours) | 36 | 3,580 | 1,122 |

Table 2: Number of feasibility violations incurred by the N-3 SCOPF version of our method, a baseline optimal power flow (OPF), and the PowerModels N-1 SCOPF solver [50] on randomly selected N-1, N-2, N-3 contingency scenarios for a 4622-node test case. We see that CAN∂Y reduces the number of N-2 and N-3 violations by a factor of 3-4× over the OPF baseline, and the number of N-3 violations by a factor of 5× over the PowerModels solution.

We find that our score is comparable against the top approaches submitted to the GO Competition, as shown in Table 1. We note that these score comparisons are not exact, as our power flow solver uses a more realistic model for coupling between the base and contingency cases than was posed in Challenge 1, which affects the way that the evaluation metric is computed (see Appendix D). Nonetheless, at a high level, these results demonstrate that our method performs competitively with respect to the top-performing methods for solving N-1 SCOPF.

## 5.3 Improving N-3 SCOPF

We now describe the performance of our method on our main setting of interest: N-3 SCOPF. While other competitive methods exist for solving N-1 SCOPF, previous work has struggled to approach settings allowing for larger numbers of simultaneous outages (e.g., N-k SCOPF for $k = 2$ or 3) due to the associated combinatorial explosion in problem size. Our method, however, scales gracefully with respect to the number of allowable simultaneous outages, as we need only tweak the value of $k$ used within our attack set $\mathcal{Y} := \{y : y \in [0, 1]^{n_o}, \|y\|_1 \leq k\}$. More specifically, each iteration of the attack maximization and each defense step calculation take approximately the same amount of time *regardless of the value of* $k$, given that the costs of the gradient computations, projections, and (optimal) power flow solves are independent of $k$. (The total number of iterations it takes for our method to converge may vary between settings, though we do not notice a substantive difference in this respect between the N-1, N-2, and N-3 versions of our approach during our experiments.)

We use our method to attempt to solve N-3 SCOPF (i.e., set $k = 3$) on a 4622-node test case over all 6133 potential outages (i.e., over 38 billion N-3 contingency scenarios); the associated training curve is shown in Figure 2. In total, our approach takes only 21 minutes to converge.

We evaluate the strength of our obtained dispatch in maintaining feasibility against a combination of N-1, N-2, and N-3 contingency scenarios, which are all technically contained within the threat model represented by our choice of $\mathcal{Y}$. We note that while full security against all these contingencies

is likely impossible with a single dispatch – e.g., we can very often construct an N-3 contingency that isolates, or *islands*, some non-self-sustaining part of the electrical grid – we aim to demonstrate that our method can improve upon existing methods in terms of providing robustness against a wide variety of scenarios. As there remain a lack of available N-2 or N-3 SCOPF methods against which we can readily compare, we compare our performance against that of a base case optimal power flow (OPF) solver, as well as the open-source PowerModels N-1 SCOPF algorithm [50]. Due to the intractability of evaluating these dispatches on *all* possible contingency scenarios, we randomly sub-select the set of N-1, N-2, and N-3 scenarios on which we evaluate, and run these evaluations in PowerWorld over the course of several days.

The results of our evaluation are shown in Table 2. Overall, we see that CAN$\partial$Y significantly reduces the number of total contingency violations as compared to the OPF solution by a factor of 3-4$\times$. While the PowerModels baseline and our approach perform comparably with respect to N-1 and N-2 contingency violations, our approach incurs nearly 5$\times$ fewer N-3 violations as compared to PowerModels. Analyzing the N-3 contingencies in more detail, we find that of the 4086 specific violations incurred by OPF, 931 of those were also incurred by CAN$\partial$Y (while the remaining 191 violations incurred by our method were disjoint). Overall, these results indicate that our method is much more effective than OPF and a benchmark N-1 SCOPF solver at guarding against N-2 and N-3 contingencies, though the actual distribution of specific contingencies that are guarded against may differ between these methods.

## 6   Conclusion

In this paper, we have described our approach, CAN$\partial$Y, for N-k security-constrained optimal power flow. Specifically, we formulate N-k SCOPF as a minimax problem over power system dispatches and potential outages by forming a continuous outer approximation to the contingency space. This enables us to compute "worst-case" contingencies via projected gradient descent (using tricks from the implicit layers literature) and then employ these contingencies to update our proposed dispatch. Notably, our formulation scales gracefully in the number of contingencies – requiring only a minor tweak to the projection set during the attack step – even as the underlying problem scales combinatorially. In particular, our approach takes only 21 minutes to converge on a standard laptop for a realistic 4622-node test case with over 38 billion potential N-3 contingencies. We show that our approach reduces N-3 feasibility violations by a factor of 3-4$\times$ compared to a baseline optimal power flow method and by almost 5$\times$ compared to a baseline N-1 SCOPF solver. Overall, we believe this demonstrates the promise of our approach in enabling scalable N-k security-constrained optimization on realistic-scale power grids.

We note that the success of our minimax optimization approach is likely highly reliant on the strength of the adversarial attacks that we generate, just as in the adversarially robust training literature [3, 4]. As such, a fruitful direction for future work may involve developing improved procedures for obtaining adversarial attacks in the context of SCOPF. In addition, given the large scale of the power networks we consider, it is generally impossible to evaluate proposed dispatches against the full suite of potential contingencies in order to check whether they are indeed N-k secure. Given that, another fruitful direction may entail developing better evaluation metrics or verification procedures to inexpensively evaluate whether a proposed SCOPF dispatch is (likely) robust, perhaps again drawing inspiration from the literature on verification methods for adversarially robust deep learning [51].

## 7   Broader impacts

To address climate and sustainability goals, many power grids are starting to integrate larger amounts of time-varying renewable energy, such as solar and wind. As described in [52], this means power systems optimization problems must generally be solved both more quickly and at larger scales. In addition, weather extremes driven by climate change [37] yield a significant need for resilient power system optimization. We believe our work makes an important contribution to this area by providing a scalable and efficient method to address N-k SCOPF. One potential concern, however, is that our method relies on identifying "worst-case" power system attacks, and our associated approach could theoretically be exploited by adversarial actors; that said, we believe it unlikely that our method in particular would be used for this purpose, as it employs a continuous "partial outage" approximation that is useful for the purposes of our algorithm, but does not map neatly to actual real-world attacks.

## Acknowledgments and Disclosure of Funding

This work was supported by the U.S. Department of Energy Computational Science Graduate Fellowship (DE-FG02-97ER25308), the Center for Climate and Energy Decision Making through a cooperative agreement between the National Science Foundation and Carnegie Mellon University (SES-00949710), the National Science Foundation (ECCS-1800812), the Computational Sustainability Network, and the Bosch Center for AI.

We thank Eric Wong, George Haff, Marko Jereminov, Amritanshu Pandey, and anonymous reviewers for their input on this work.

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
