## A  Full SCOPF formulation

In the N-k SCOPF formulation presented in Equation (6), for brevity, we abstracted away various device constraints and operational limits into the sets $\mathcal{X}, \mathcal{W}_{\text{base}}, \mathcal{Z}_i$, and $\mathcal{W}_i$. Here, we write out those constraints more explicitly to facilitate in-depth descriptions of components of our attack and defense in Appendices B and C.

Specifically, let $n_{\text{bus}}$ be the number of nodes (*buses*) on the power system, and let $n_g$ be the number of generators. By convention, we designate one of these generators to be a *slack bus* whose voltage angle is fixed at a particular value. Our dispatch $x \in \mathbb{R}^{2n_g - 1}$ then consists of the voltage magnitude at the slack bus, and the real power generation and voltage magnitude at all other generators, subject to device limits and operational constraints. As in Section 4.1, we let $\mathcal{C}$ represent the set of contingencies, and $z^{(i)} \in \mathbb{R}^{2n_g - 1}$ represent slightly adjusted settings of the dispatch quantities that the power system operator can create after scheduling $x$ and then observing some contingency $c^{(i)} \in \mathcal{C}$. We then write the N-k SCOPF problem as

$$
\begin{aligned}
\underset{x \in \mathbb{R}^{2n_g-1}}{\text{minimize}} \quad & f_{\text{base}}(x) + \sum_{(z^{(i)}, c^{(i)})} f_{\text{cont}}(z^{(i)}, c^{(i)}) \\
\text{subject to} \quad & g_{\text{flow,base}}(x, w_{\text{base}}) = 0, \quad h_{\text{base}}(x, w_{\text{base}}) \le 0, \quad w_{\text{base}} \in \mathbb{R}^d \\
& z^{(i)} \in 
\begin{array}{l}
\text{argmin}_{z^{(i)} \in \mathbb{R}^{2n_g-1}} f_{\text{cont}}(z^{(i)}, c^{(i)}) \\
\text{s.t. } g_{\text{flow,cont}}(z^{(i)}, w^{(i)}, x) = 0 \\
\quad\quad h_{\text{cont}}(z^{(i)}, w^{(i)}, x, c^{(i)}) \le 0 \\
\quad\quad w^{(i)} \in \mathbb{R}^d
\end{array}
\quad \forall c^{(i)} \in \mathcal{C},
\end{aligned} \tag{A.1}
$$

where $h_{\text{base}} : \mathbb{R}^{2n_g-1} \times \mathbb{R}^d \to \mathbb{R}^{2 \times (2n_g-1+d)}$ represents device and operational constraints (box constraints) on $x$ and $w_{\text{base}}$, $h_{\text{cont}} : \mathbb{R}^{2n_g-1} \times \mathbb{R}^d \times \mathbb{R}^{2n_g-1} \times \mathcal{C} \to \mathbb{R}^{2 \times (2n_g-1+d)}$ represents device and operational constraints (box constraints) on $z^{(i)}$ and $w^{(i)}$, and all other quantities are as defined in Section 4.1.

We can then re-write our N-k SCOPF minimax formulation presented in Section 4.2 as

$$
\underset{x \in \mathbb{R}^{2n_g-1}}{\text{minimize}} \max_{y \in \mathcal{Y}} \quad f_{\text{base}}(x) + f_{\text{cont}}(z, y) + \frac{1}{2}\|s\|_2^2 \tag{A.2a}
$$

$$
\text{subject to} \quad g_{\text{flow,base}}(x, w_{\text{base}}) = 0, \quad h_{\text{base}}(x, w_{\text{base}}) \le 0, \quad w_{\text{base}} \in \mathbb{R}^d \tag{A.2b}
$$

$$
z, s \in 
\begin{array}{l}
\text{argmin}_{z \in \mathbb{R}^{2n_g-1}, \, s \in \mathbb{R}^{2n_{\text{bus}}}} f_{\text{cont}}(z, y) + \frac{1}{2}\|s\|_2^2 \\
\text{s.t. } g_{\text{flow,cont}}(z, w_{\text{cont}}, x) + s = 0 \\
\quad\quad h_{\text{cont}}(z, w_{\text{cont}}, x, y) \le 0 \\
\quad\quad w_{\text{cont}} \in \mathbb{R}^d,
\end{array} \tag{A.2c}
$$

where $\mathcal{Y} := \{y : y \in [0,1]^{n_o}, \|y\|_1 \le k\}$ is our outer relaxation to the contingency set, and $s \in \mathbb{R}^{2n_{\text{bus}}}$ are slack variables representing potential infeasibilities in the third-stage optimization problem.

## B  Further details on the SCOPF attack

The innermost optimization problem in Equation (A.2c), represents the decision a power grid operator would make when faced with a particular partial contingency $y^\star$ after having made a base dispatch $\bar{x}$. In particular, the grid operator is looking to minimize the cost of power generation $f_{\text{cont}}(z, y^\star)$, as well as ensure the system remains feasible ($s = 0$). To solve for the grid response due to a particular partial contingency, we solve (A.2c) using a Newton-based approach [45].

Specifically, let $\lambda \in \mathbb{R}^{2n_{\text{bus}}}$ be the dual variables on the power flow constraint, and $\mu \in \mathbb{R}^{2 \times (2n_g-1+d)}$ be the dual variables on the inequality constraints. The Lagrangian of the optimization problem is given by

$$
\mathcal{L} = f_{\text{cont}}(z, y^\star) + \frac{1}{2}\|s\|_2^2 + \lambda^T \left( g_{\text{flow,cont}}(z, w_{\text{cont}}, \bar{x}) + s \right) + \mu^T h_{\text{cont}}(z, w_{\text{cont}}, \bar{x}, y^\star). \tag{B.1}
$$

The KKT conditions for stationarity, primal feasibility, complementary slackness, and dual feasibility are then given by

$$
\frac{\partial \mathcal{L}}{\partial z} = \frac{\partial f_{\text{cont}}(z, y^\star)}{\partial z} + \left( \frac{\partial g_{\text{flow,cont}}(z, w_{\text{cont}}, \bar{x})}{\partial z} \right)^T \lambda + \left( \frac{\partial h_{\text{cont}}(z, w_{\text{cont}}, \bar{x}, y^\star)}{\partial z} \right)^T \mu = 0
$$

$$
\frac{\partial \mathcal{L}}{\partial s} = s + \lambda = 0
$$

$$
g_{\text{flow,cont}}(z, w_{\text{cont}}, \bar{x}) + s = 0 \tag{B.2}
$$

$$
\text{diag}(\mu) h_{\text{cont}}(z, w_{\text{cont}}, \bar{x}, y^\star) = 0
$$

$$
\mu \geq 0.
$$

The KKT conditions can be written as a set of equations $F_{\text{attack}}(z, s, \lambda, \mu) = 0$. Traditionally in the power systems literature, the equations in (B.2) are solved using a Newton's method [45]. The iterative Newton's method starts with some initial estimates $z^0, s^0, \lambda^0, \mu^0$ for $z, s, \lambda, \mu$. Then, at each iteration $i$, we construct a Jacobian $J^i$ for $F_{\text{attack}}$ and a corresponding right hand side vector $b^i$ at our current estimate $z^{i-1}, s^{i-1}, \lambda^{i-1}, \mu^{i-1}$. We then solve the resultant set of linear equations

$$
J^i \begin{pmatrix} z^i \\ s^i \\ \lambda^i \\ \mu^i \end{pmatrix} = b^i \tag{B.3}
$$

to determine the next estimate for the solution, and iterate until convergence.

## C  Further details on the SCOPF defense

The defense step of our approach entails adjusting our dispatch in a direction of increased robustness to the "worst-case" contingency $y^\star$ found in the attack stage, while also maintaining feasibility in the base case. To do so, we partially solve the minimization problem shown in Equation (12), as described in the main text. Specifically, let $\lambda_{\text{base}}$ and $\lambda_{\text{cont}}$ denote the dual variables on the power flow constraints in the base and contingency cases, respectively, and let $\mu_{\text{base}}$ and $\mu_{\text{cont}}$ denote the dual variables on the inequality constraints in the base and contingency cases, respectively. Then, the Lagrangian of optimization problem (12) is given by

$$
\begin{aligned}
\mathcal{L} = f_{\text{base}}(x) + f_{\text{cont}}(z, y^\star) + \frac{1}{2}\|s\|_2^2 + \lambda_{\text{base}}^T g_{\text{flow,base}}(x, w_{\text{base}}) \\
+ \lambda_{\text{cont}}^T(g_{\text{flow,cont}}(z, w_{\text{cont}}, x) + s) + \mu_{\text{base}}^T h_{\text{base}}(x, w_{\text{base}}) + \mu_{\text{cont}}^T h_{\text{cont}}(z, w_{\text{cont}}, x, y^\star).
\end{aligned} \tag{C.1}
$$

The KKT conditions associated with this problem are given by

$$
\begin{aligned}
\frac{d\mathcal{L}}{dx} = \frac{\partial f_{\text{base}}(x)}{\partial x} + \left( \frac{\partial g_{\text{flow,base}}(x, w_{\text{base}})}{\partial x} \right)^T \lambda_{\text{base}} + \left( \frac{\partial h_{\text{base}}(x, w_{\text{base}})}{\partial x} \right)^T \mu_{\text{base}} \\
+ \left( \frac{\partial g_{\text{flow,cont}}(z, w_{\text{cont}}, x)}{\partial x} \right)^T \lambda_{\text{cont}} + \left( \frac{\partial h_{\text{cont}}(z, w_{\text{cont}}, x, y^\star)}{\partial x} \right)^T \mu_{\text{cont}} = 0
\end{aligned} \tag{C.2}
$$

$$
\frac{d\mathcal{L}}{d\lambda_{\text{base}}} = g_{\text{flow,base}}(x, w_{\text{base}}) = 0 \tag{C.3}
$$

$$
\text{diag}(\mu_{\text{base}}) h_{\text{base}}(x, w_{\text{base}}) = 0 \tag{C.4}
$$

$$
\frac{d\mathcal{L}}{dz} = \frac{\partial f_{\text{cont}}(z, y^\star)}{\partial z} + \left( \frac{\partial g_{\text{flow,cont}}(z, w_{\text{cont}}, x)}{\partial z} \right)^T \overset{0}{\cancel{\lambda_{\text{cont}}}} + \left( \frac{\partial h_{\text{cont}}(z, w_{\text{cont}}, x, y^\star)}{\partial z} \right)^T \overset{0}{\cancel{\mu_{\text{cont}}}} = 0 \tag{C.5}
$$

$$\frac{\mathrm{d}\mathcal{L}}{\mathrm{d}s} = s + \lambda_{\text{cont}} = 0 \tag{C.6}$$

$$\frac{\mathrm{d}\mathcal{L}}{\mathrm{d}\lambda_{\text{cont}}} = g_{\text{flow,cont}}(z, w_{\text{cont}}, x) + s = 0 \tag{C.7}$$

$$\text{diag}(\mu_{\text{cont}})h_{\text{cont}}(z, w_{\text{cont}}, x, y^\star) = 0, \tag{C.8}$$

where a number of the terms in condition (C.5) cancel to zero due to the underlying structure of N-k SCOPF problem. We can group the KKT conditions together based on their relation to the base variables or the contingency variables. We define two vectors of equations that group the KKT conditions together: $F_{\text{base}}(x, w_{\text{base}}, \lambda_{\text{base}}, \mu_{\text{base}})$, representing the decoupled defense equations in blue above, and $F_{\text{cont}}(z, w_{\text{cont}}, \lambda_{\text{cont}}, \mu_{\text{cont}})$, representing the decoupled contingency optimization equations in red. We can then group the KKT conditions as follows:

$$\begin{pmatrix} \frac{\mathrm{d}\mathcal{L}}{\mathrm{d}x} \\ \frac{\mathrm{d}\mathcal{L}}{\mathrm{d}\lambda_{\text{base}}} \\ \text{diag}(\mu_{\text{base}})h_{\text{base}}(x, w_{\text{base}}) \end{pmatrix}$$

$$\equiv F_{\text{base}}(x, w_{\text{base}}, \lambda_{\text{base}}, \mu_{\text{base}}) + \begin{pmatrix} \left(\frac{\partial g_{\text{flow,cont}}(z, w_{\text{cont}}, x)}{\partial x}\right)^T \lambda_{\text{cont}} + \left(\frac{\partial h_{\text{cont}}(z, w_{\text{cont}}, x, y^\star)}{\partial x}\right)^T \mu_{\text{cont}} \\ 0 \\ 0 \end{pmatrix} = 0, \tag{C.9}$$

$$\begin{pmatrix} \frac{\mathrm{d}\mathcal{L}}{\mathrm{d}z} \\ \frac{\mathrm{d}\mathcal{L}}{\mathrm{d}s} \\ \frac{\mathrm{d}\mathcal{L}}{\mathrm{d}\lambda_{\text{cont}}} \\ \text{diag}(\mu_{\text{cont}})h_{\text{cont}}(z, w_{\text{cont}}, x, y^\star) \end{pmatrix} \tag{C.10}$$

$$\equiv F_{\text{cont}}(z, w_{\text{cont}}, \lambda_{\text{cont}}, \mu_{\text{cont}}) + \begin{pmatrix} 0 \\ 0 \\ g_{\text{flow,cont}}(z, w_{\text{cont}}, x) \\ \text{diag}(\mu_{\text{cont}})h_{\text{cont}}(z, w_{\text{cont}}, x, y^\star) \end{pmatrix} = 0.$$

We notice the two KKT terms $F_{\text{base}}$ and $F_{\text{cont}}$ are independent but are coupled through two additional terms. For the N-k SCOPF application, these coupling terms represent the generator's contingency ramping constraints and the voltage set points from the base case. Due to the sparse nature of the grid, these couplings are weak, which makes these equations well-suited to solve using a decoupled Gauss-Seidel approach.

The non-linear Gauss-Seidel is an iterative method to solve the two sets of weakly coupled equations independently using the values of the coupled variables from the previous iteration. Based on the KKT conditions above, we can write the Gauss-Siedel equations at iteration $i$ as follows:

$$F_{\text{cont}}(z^i, w_{\text{cont}}^i, \lambda_{\text{cont}}^i) + \begin{pmatrix} 0 \\ 0 \\ g_{\text{flow,cont}}(z^i, w_{\text{cont}}^i, x^{i-1}) \\ \text{diag}(\mu_{\text{cont}}^i)h_{\text{cont}}(z^i, w_{\text{cont}}^i, x^{i-1}, y^\star) \end{pmatrix} = 0, \tag{C.11}$$

$$F_{\text{base}}(x^i, w_{\text{base}}^i, \lambda_{\text{base}}^i, \mu_{\text{base}}^i) + \begin{pmatrix} \frac{\partial g_{\text{flow,cont}}(z^i, w_{\text{cont}}^i, y^\star)}{\partial x}^T \lambda_{\text{cont}}^i + \left(\frac{\partial h_{\text{cont}}(z^i, w_{\text{cont}}^i, x^i, y^\star)}{\partial x}\right)^T \mu_{\text{cont}}^i \\ 0 \\ 0 \end{pmatrix} = 0. \tag{C.12}$$

By specifically ordering the Gauss-Seidel algorithm to first solve (C.11) and then (C.12), we can pass the updated contingency-specific coupling variables $z^i$ and $\lambda^i_{\text{cont}}$ obtained by solving (C.11) at the current iterate $i$ to the solve of (C.12). The Gauss-Seidel algorithm for this specific application relaxes the coupling by, at iteration $i$, using the value of $x$ from the previous iteration, as highlighted in green in (C.11). By ordering the updates in this way, we initiate the Gauss-Seidel iterations using the final solution we already obtained when solving for the worst-case attack $y^\star$ and its associated $z^\star$, which is the solution to (13b).

Rather than running the Gauss-Seidel iterations to convergence, we run only a single iteration in order to take a step in the dispatch $x$ (before then restarting the attack phase). We can efficiently solve this single Gauss-Seidel since we reuse the solution from the attack stage for Equation (C.11), and therefore all that remains to solve is Equation (C.12).

# D  GO competition scoring

Challenge 1 of the GO competition [34] created a specific formulation for the grid constraints in Equation (A.2c). In particular, they relaxed part of the formulation to not allow certain grid models such as *switched shunts*. They also used *automatic generation control* modulate the power generation at each generator to ensure the power grid frequency remained at its nominal set point. However, in contrast, our "third stage" power flow solver is built to solve discrete shunts and discrete *transformer tap controls*. We also do not use automatic generation control, but instead enable generators to *ramp* (i.e., change their production) within some limits determined by the base dispatch. These grid models are most consistent with the ongoing Challenge 2 of the GO competition [34], which has updated its grid models since the first competition. The full list of relevant differences in grid models is shown in the table below.

|  | GO Challenge 1 models | CAN$\partial$Y models |
| --- | --- | --- |
| Power generation adjustments | Automatic generation control | Generator ramping |
| Discrete shunts | Not allowed | Allowed |
| Transformer tap ratios | Fixed | Adjustable (discrete) |

As scores and solutions for GO Challenge 2 are not yet available, we compare our solver against solvers from GO Challenge 1, despite the fact that our solver uses grid models from GO Challenge 2. In particular, we score our formulation by using the score weighting criteria given in GO Challenge 1 [34] and compare the against the scores of the top Challenge 1 competitors, which are available at [34]. However, since the grid models used by our solver vs. the Challenge 1 solvers are different, the score comparisons we show in Section 5.2 are ultimately approximate. Nonetheless, we believe that the comparisons still indicate that our methodology is competitive for N-1 SCOPF.