# OpenReview forum: "Adversarially robust learning for security-constrained optimal power flow"
_NeurIPS.cc/2021/Conference — NeurIPS 2021 Poster_

### Official Review · Reviewer_MetJ · 2021-07-13

**Rating:** 8
**Confidence:** 4

**Summary:**

The authors develop a new method for solving the N-k SCOPF problem, an optimal power flow problem robust to any $k$ outages. Specifically, they relax the problem to allow for any convex combination of outages that has total 1-norm less than $k$. They then leverage recently popularized implicit differentiation techniques to solve the resulting minimax problem. Their formulation works for generic minimax problems, although one can exploit problem-specific structure when solving the N-k SCOPF problem. Finally, they test their method on a reasonably large test case for $k = 1, 2, 3$, and show that it finds a solution in a reasonable amount of time that improves upon standard OPF (in terms of the number of violations).

**Limitations And Societal Impact:**

Although the author allude to the lack of theoretical guarantees associated with their method in the first paragraph of the introduction, it would be nice to state this limitation explicitly later in the paper.

The broader impact, including the potential negative societal impact, are clearly stated in the paper.

**Main Review:**

The main contribution of the paper is to use implicit differentiation to solve a relaxed version of the N-k SCOPF problem. This contribution is novel (as far as I’m aware of) and clearly addresses the scalability issues that arise when solving robust OPF problems on large networks. Given the large size of most real power networks, the paper represents an important step towards scaling N-k OPF to real networks (for $k >= 2$ of course).

The results appear technically sound, and the writing is very clear. One statement I found slightly confusing was that on line 223—why wouldn’t projected gradient descent guarantee feasibility (shouldn’t the projection onto the feasible set take care of this)?

The experiments section clearly verified that the method works and produces reasonable results in an acceptable amount of time. However, I’m not 100% certain this method produces better results than a simple robust OPF solver (how does it compare to N-1 SCOPF), since comparing directly to standard OPF is a bit of a straw man. Would it be possible to add the number of violations produced by N-1 SCOPF to Table 2? This would give the reader a better sense of how this method compares to standard robust OPF methods (especially since N-1 is so often used in practice).



Minor Comments
- On line 95, should maximizer and minimizer be switched?
- What is the origin / purpose / utility of the name CAN$\partial$Y?
- In Table 1, is a lower or higher score better?

**Time Spent Reviewing:**

5

---

> ### Author Response · Authors · 2021-08-10
> **Response to Reviewer MetJ**
>
> Thank you for your interest in our work and for the helpful feedback!
>
> To address some specific comments raised within your review:
>
> > One statement I found slightly confusing was that on line 223—why wouldn’t projected gradient descent guarantee feasibility (shouldn’t the projection onto the feasible set take care of this)?
>
> Thanks for pointing this out; we were unclear in our description here. What we meant to say is that projected gradient descent with respect to the set $\mathcal{X}$ (representing generator-wise box constraints on the dispatch) would not enforce feasibility of the power flow equations $g_{\text{flow}}(\cdot) = 0$. You are correct that an alternative approach would be to project onto both $\mathcal{X}$ and $g_{\text{flow}}(\cdot) = 0$. However, since $g_{\text{flow}}$ is (worse than) quadratic, this represents a projection onto a nonconvex set of constraints that is in practice quite expensive to solve. As a result, our method provides a cheaper alternative. We will be sure to make this point much clearer in the next iteration of our paper.
>
> > However, I’m not 100% certain this method produces better results than a simple robust OPF solver (how does it compare to N-1 SCOPF), since comparing directly to standard OPF is a bit of a straw man. Would it be possible to add the number of violations produced by N-1 SCOPF to Table 2?
>
> We certainly agree that comparing to a baseline N-1 SCOPF solver in Table 2 would be ideal. One challenge we experienced, however, was that we were not able to easily obtain access to such a baseline solver,* since e.g. the code for these solvers is not often open source. This is why we took a two-pronged comparison approach in our experiments, namely (1) comparing against the scores obtained by other N-1 SCOPF solvers in the GO competition setting, and (2) comparing against a state-of-the-art OPF solver in N-k settings.
>
> [*] Note: Our work considers the setting of N-k SC-ACOPF, i.e., N-k SCOPF on the true nonlinear physical model of the grid. We were not able to obtain access to an N-1 SC-ACOPF baseline solver. While we do have access to an N-1 SCOPF solver that uses a linear approximation of the grid (i.e., addresses the N-1 SC-DCOPF setting), the solutions of these systems are generally not feasible in the true nonlinear system [1]. Thus, we did not think this would be a fair comparison.
>
> [1] Baker, Kyri. "Solutions of DC OPF are never AC feasible." Proceedings of the Twelfth ACM International Conference on Future Energy Systems. 2021.
>
> > ​​On line 95, should maximizer and minimizer be switched?
>
> Yes! Apologies for the typo.
>
> > What is the origin / purpose / utility of the name CAN∂Y?
>
> We’ll plan on explaining this in the final version of the paper, but wanted to abstain from doing so in this version, as it could potentially deanonymize the submission.
>
> > In Table 1, is a lower or higher score better?
>
> Lower is better; we’ll plan to update the caption to make this clearer.

---

> > ### Comment · Reviewer_MetJ · 2021-08-10
> > **Baseline algorithm**
> >
> > Here is a publicly available baseline algorithm. It is the benchmark in the ARPA-E competition. Based on the slides (link attached), I believe it solves AC SCOPF (not DC SCOPF).
> >
> > https://github.com/lanl-ansi/PowerModelsSecurityConstrained.jl
> > https://arpa-e.energy.gov/sites/default/files/Benchmark%20-%20Carleton%20Coffrin.pdf

---

> > > ### Author Response · Authors · 2021-08-13
> > > **Will update on baseline algorithm**
> > >
> > > Thank you for the pointer. We're currently working on running this and will report back with results.

---

> > > > ### Author Response · Authors · 2021-08-17
> > > > **Update on baseline comparisons**
> > > >
> > > > Thanks again for the pointer. To share some preliminary results for the N-2 SCOPF setting, out of 56,468 scenarios we have tested so far, here are the results:
> > > > * Base OPF: 1936 violations
> > > > * PowerModels SCOPF: 169 violations
> > > > * Our method: 94 violations
> > > >
> > > > So, based on our results so far, our method does substantially outperform at least the PowerModels SCOPF algorithm (with an over 40% reduction in violations) on the N-2 SCOPF setting.
> > > >
> > > > Please note that this is only a subset of the N-2 SCOPF scenarios tested out in the paper, and we are in the process of testing out the full number of scenarios for both N-2 and N-3. However, since these tests take a while to run, we wanted to report back with some initial results (and will of course plan on incorporating the full results into the paper).

---

> > > > > ### Comment · Reviewer_MetJ · 2021-08-17
> > > > > **Baseline**
> > > > >
> > > > > Great! I realize my request was highly non-trivial for the very limited revision period, and I appreciate the authors taking the time to carry this out.
> > > > >
> > > > > Since the proposed method has now been compared to a very reasonable benchmark, I will raise my score to an 8.

---

### Official Review · Reviewer_eZgw · 2021-07-16

**Rating:** 6
**Confidence:** 4

**Summary:**

In this paper, N-k SCOPF is formulated and solved through the lens of adversarial learning and the implicit function theory. Thus, problem formulation and the development of a multi-level optimization algorithm are two key contributions. Meanwhile, the paper is mostly well written and easy to follow.

**Limitations And Societal Impact:**

Yes, the authors have adequately discussed the limitations and potential negative societal impact.

**Main Review:**

Strength:

+ Rewriting N-k SCOPF as a minimax problem is a novel idea.

+ Obtaining (implicit) attack gradients via KKT conditions is interesting.

+ The proposed seems scalable even if high-order derivatives and matrix inversion exist.

Weakness:

- Although the paper is written well in general, several important details are missing in the main paper. For example, Appendix B should be placed in the main body since this is a critical derivation to obtain implicit gradients. Moreover, a pseudo-algorithm that summarizes both attacks and defend steps will make the proposed optimization steps clear.

- In Eq. (11), it is not clear how to simplify the computation of $J^{-1}$.  Also, will $\frac{d J}{d y}$ involve higher-order derivatives? If so, how to overcome the scalability issue?

- A computation complexity analysis of the proposed algorithm, e..g, in the big O notation, will make the scalability argument more convincing.

- The implementation details of the proposed algorithm were not clearly discussed in experiments. E.g., hyper-parameter selection associated with the algorithm?


**Time Spent Reviewing:**

3 hours

---

> ### Author Response · Authors · 2021-08-10
> **Response to Reviewer eZgw [part 1/2]**
>
> Thank you for your review and for the helpful feedback. From our perspective, your comments seem to fall into two categories: (a) questions about the scalability of Jacobian-related computations, and (b) suggestions for improving the writing or presentation. We address both categories of comments below.
>
> ## Scalability of Jacobian-related computations
>
> In your review, you correctly point out that efficiently computing $J^{-1}$ and $\frac{\mathrm{d} J}{\mathrm{d} y}$ is critical to the efficiency of the overall method. To provide some context, the fundamental reason we are able to compute these quantities efficiently is due to the specialized structure of the Jacobian matrix $J$.
>
> Specifically, we compute $J$ within the loop of an efficient power flow solver, which employs a power system model based on circuit simulation. Notably, under this circuit simulation model, $J$ is extremely sparse. For example, a mid-sized (5000-node) power network has a Jacobian $J$ that is 99.92% sparse, and this level of sparsity is roughly consistent with larger networks as well. In addition, $\frac{\mathrm{d} J}{\mathrm{d} y}$ is even more sparse, with less than 16 nonzero entries per dimension of $y$. This is due to the particular properties of the circuit simulation models we use.
>
> In the next iteration of our submission, we will be sure to more explicitly emphasize the sparsity of the Jacobian-related matrices, as this is indeed one major reason that our algorithm is so fast. While we wanted to avoid explicit citations to our power flow solver (which we developed in previous work) in order to preserve anonymity, we will also provide these citations in the deanonymized version of our submission in order to provide additional context on why $J$ and $\frac{\mathrm{d} J}{\mathrm{d} y}$ are so sparse.
>
> To address your specific comments:
>
> > In Eq. (11), it is not clear how to simplify the computation of $J^{−1}$.
>
> The Newton power flow solver that we use in the forward pass employs sparse LU factorization-based tricks to efficiently compute $J^{-1}$ as part of the power flow solve. In the backward pass — i.e., the gradient computation in Equation (11) — we simply adopt the value of $J^{-1}$ computed during the forward pass. We will plan to state this more explicitly, with a citation to the prior work, in the next iteration of the paper.
>
> To clarify further, however, the description after Equation (11) pertains not to simplifying the computation of $J^{-1}$, but instead to efficiently computing the gradient of the loss, $\frac{\mathrm{d} \ell}{\mathrm{d} y}$. The main ideas here are that we can (a) reuse the value of $J^{-1}$ obtained during the forward pass (as described above) and (b) intelligently order our matrix multiplications in a way that reduces the overall computational complexity of these matrix multiplications.
>
> > Also, will $\frac{dJ}{dy}$ involve higher-order derivatives? If so, how to overcome the scalability issue?
>
> Great question. $\frac{\mathrm{d} J}{\mathrm{d} y}$ does involve higher-order gradients, but these are easy to compute in our setting for two main reasons:
> * $\frac{\mathrm{d} J}{\mathrm{d} y}$ is extremely sparse: As defined in the paper, let $n_o$ be the number of generators or power lines that can potentially experience an outage (i.e., $y \in [0, 1]^{n_o}$). As mentioned above, the number of nonzero entries in $\frac{\mathrm{d} J}{\mathrm{d} y}$ is then at most $16 \times n_o$. This is because $J$ represents the linearization of four different circuits (primal and dual circuits, both real and imaginary), each of which has four terms governing each generator or line. Thus, each entry in $y$ directly affects at most $4 \times 4 = 16$ terms in $J$.
> * The few terms that are nonzero have nice analytical forms, and we can actually reuse derivatives obtained during the power flow solve in order to get these terms basically “for free.”
>
> > A computation complexity analysis of the proposed algorithm, e..g, in the big O notation, will make the scalability argument more convincing.
>
> While the exact computational complexity of each step relies on the sparsity structure of $J$, and thus depends on the exact structure of each power network we address, the complexity at a high level for a system with $n$ nodes and potentially $k$ simultaneous outages is as follows:
>
> **Attack Stage:** The attack stage, described in Section 4.3, uses a power flow solver (using Newton Raphson) and projected gradient descent to find a contingency of maximal impact.
>
> The total complexity for the attack stage for $m$ attack iterations is
> $$m \times (O(tn^3) + O(n^2) + O(k)),$$
> where $t$ is the number of Newton-Raphson iterations for the power flow solve in the forward pass, $O(t n^3)$ is thus an upper bound on the complexity of this forward pass, and $O(n^2) + O(k)$ describes the computational complexity of the backward pass. As we note below, $O(tn^3)$ is actually a strict overestimate for the complexity of the power flow solver (developed outside the context of this work), as in practice that solver exploits the sparsity structure of the network to achieve much better complexity in practice.
>
> To describe this in more detail:
> * *Power flow solve (forward pass):* The complexity for the power flow solver is generally dominated by a linear solve. A naive implementation of this linear solve would achieve $O(n^3)$ complexity; given $t$ iterations for the Newton-Raphson to converge, we would then expect an upper bound for the power flow solver of $O(tn^3)$. However, the power flow solver we use achieves much faster performance in practice due to the sparsity of the network. In addition, this solver uses circuit heuristics to minimize the number of iterations, $t$.
> * *Projected gradient step (backward pass):* After solving the power flow, the projected gradient step calculates relevant attack gradients in (11). By ordering the gradient calculation as described in 4.3, and exploiting the sparsity structure of our power network, we reduce the complexity of this gradient computation to being dominated by a single $O(n^2)$ matrix-vector product.
>     * Using the “vector-Jacobian product trick”, we solve for the gradients from left to right, thereby always computing a vector-matrix product. Importantly, we save the Jacobian inverse from the previous Newton-Raphson solve, and avoid re-inverting the matrix. The first vector-matrix product in the blue in equation (11) results in an $O(n^2)$ complexity.
>     * The next step in calculating the attack gradient is to multiply the resulting vector with $-\frac{dJ}{dy}\begin{pmatrix} z^\star \\\\ s^\star \end{pmatrix}+\frac{db}{dy}$. We exploit the sparsity of the power network in order to efficiently compute this vector-matrix computation. Intuitively, for a single device under contingency with a continuous contingency parameter, $y_i$, the vector $\frac{dJ}{dy_i}\begin{pmatrix} z^\star \\\\ s^\star \end{pmatrix}$ results in a sparse matrix with only few non-zero rows related to the location of the actual device. With this knowledge, a manual computation of this vector-vector product results in $O(1)$ complexity for each contingency. With $k$ possible contingencies, the complexity of this final vector-vector computation is $O(k)$.
>
> **Defense Stage:** At each defense step, we solve a single iteration of a coupled base and worst case contingency system, utilizing the solution from the final attack. The complexity of this process is equivalent to solving a Newton-Raphson based AC optimal power flow for the amended base case, which for $t_{\text{defense}}$ Newton-Raphson iterations, has a complexity of $O(t_{\text{defense}} n^3)$ per defense iteration.

---

> > ### Author Response · Authors · 2021-08-10
> > **Response to Reviewer eZgw [part 2/2]**
> >
> > ## Points on writing
> >
> > > Although the paper is written well in general, several important details are missing in the main paper. For example, Appendix B should be placed in the main body since this is a critical derivation to obtain implicit gradients.
> >
> > The crux of our approach for obtaining attack gradients (Section 4.3) is that we implicitly differentiate through the final iteration of Newton’s method to obtain these gradients. To describe this, in Equation (10), we provide both the form of the Newton update at its fixed point ($J\begin{pmatrix} z^\star \\\\ s^\star\end{pmatrix}=b$) and the derivation of the implicit gradients through this update.
> >
> > We chose not to put the derivation of the Newton update itself within the main text for several reasons. The first is that we felt providing these details would actually distract from the main conceptual point: Given a power flow solver, we can differentiate through its fixed point equations. (As described above, however, we will plan on making the sparsity structure of $J$ more clear.) Second, the method for deriving Newton updates is relatively well-known and standard, rather than being a novel part of our approach. Third, a proper description of this derivation depends on the fully expanded notation for SCOPF in Appendix A, whereas we opt to provide a more simple version of the SCOPF notation in the main body of the paper to improve accessibility for an ML audience.
> >
> > In general throughout the paper, our goal was to make the main body of the paper conceptually self-contained in a way that was accessible to the three communities we address (adversarial robustness, implicit differentiation, and power systems optimization), while including additional mathematical detail in the appendix. We believe that this approach is validated by the responses of the other reviewers, who found the paper accessible despite differing areas of expertise. That said, we would of course welcome any specific additional suggestions for points that should be expanded on or moved between the main body of the paper and the appendix.
> >
> > > Moreover, a pseudo-algorithm that summarizes both attacks and defend steps will make the proposed optimization steps clear.
> >
> > Thank you for the suggestion. We will absolutely plan to include an algorithm box in the next iteration of the paper.
> >
> > > The implementation details of the proposed algorithm were not clearly discussed in experiments. E.g., hyper-parameter selection associated with the algorithm?
> >
> > The implementation is ultimately fairly simple, and we have included the code as part of the supplementary info. In particular, as we describe in item 3(b) of the checklist (line 471), the only hyperparameter we explicitly set is the convergence tolerance of the method, which we set to 1e-3 (an acceptable tolerance threshold in the settings we consider.) While we additionally set a threshold for the maximum number of inner loop and outer loop iterations, these thresholds are never actually hit. There are additionally hyperparameters associated with the power flow solver that we use, but we simply adopt the hyperparameters determined in prior work. We will plan to describe this more explicitly in the next iteration of the paper.

---

> > > ### Comment · Reviewer_eZgw · 2021-08-12
> > > **Good response**
> > >
> > > Thanks for the detailed response. My main concern on computational complexity has been addressed. Thus, I decided to increase my score.

---

### Official Review · Reviewer_h3da · 2021-07-17

**Rating:** 7
**Confidence:** 2

**Summary:**

This paper formulates a challenging optimal power flow problem as adversarial optimization problem, where the adversarial optimization problem has a challenging equality constraint.  The paper gives a principled approach to solve this problem and give convincing empirical results of the superiority of the approach.

**Limitations And Societal Impact:**

Yes

**Main Review:**

It is delightful read. The paper is very crisp: problem definitions, approach, derivations, etc. And empirical results demonstrate convincingly its effectiveness, especially for N-2, and N-3 settings.

I am however not familiar with optimal power flow problem -- Is it really important to solve N-2 and N-3 settings? More discussions would be useful.

**Time Spent Reviewing:**

2

---

> ### Author Response · Authors · 2021-08-10
> **Response to Reviewer h3da**
>
> Thank you for the kind words!
>
> Regarding the importance of N-2 and N-3 settings, these settings are indeed becoming increasingly important to solve. Some recent examples of N-k failures include major power outages in the UK in August 2019 [1] and in Texas, USA in February 2021 [2]. In addition, previous work has shown that power system failures are correlated due to factors such as temperature [3]; as temperature extremes become more common due to climate change [4], it thus becomes even more likely that failures will come in groups. Finally, in some emerging economies (including one whose power system operators we work with), having at least one outage per day is commonplace, which means contingency planning needs to account for more than one outage. Together, these factors make the development of N-k techniques extremely critical for the safe operation of power grids. We will be sure to better motivate these points in the next iteration of the paper.
>
> [1] See, e.g., https://www.current-news.co.uk/news/national-grid-eso-probing-power-cuts-following-sudden-generation-collapse
> [2] See, e.g., https://www.utilitydive.com/news/ercot-releases-plan-to-boost-reliability-after-blackouts-as-report-outline/603263/
> [3] Murphy, Sinnott J. Correlated Generator Failures and Power System Reliability. Diss. Carnegie Mellon University, 2019.
> [4] Seneviratne, S.I., N. Nicholls, D. Easterling, C.M. Goodess, S. Kanae, J. Kossin, Y. Luo, J. Marengo, K. McInnes, M. Rahimi, M. Reichstein, A. Sorteberg, C. Vera, and X. Zhang, 2012: Changes in climate extremes and their impacts on the natural physical environment. In: Managing the Risks of Extreme Events and Disasters to Advance Climate Change Adaptation [Field, C.B., V. Barros, T.F. Stocker, D. Qin, D.J. Dokken, K.L. Ebi, M.D. Mastrandrea, K.J. Mach, G.-K. Plattner, S.K. Allen, M. Tignor, and P.M. Midgley (eds.)]. A Special Report of Working Groups I and II of the Intergovernmental Panel on Climate Change (IPCC). Cambridge University Press, Cambridge, UK, and New York, NY, USA, pp. 109-230

---

### Official Review · Reviewer_SScg · 2021-07-18

**Rating:** 6
**Confidence:** 1

**Summary:**

This paper seeks to apply the techniques developed for adversarial robustness in machine learning to the problem of robust optimal power flow. It presents a formulation for the robust optimization problem, shows an efficient gradient-based optimization method and evaluates the effectiveness of the method.

**Limitations And Societal Impact:**

The authors have discussed the limitations and societal impact of the work.

**Main Review:**

I have to admit that this paper is a little outside of my expertise. My experience is mostly in adversarial robustness in machine learning. This work is more about optimal power flow. The paper does not seem to have any connection to adversarial learning apart from the fact that it uses adversarial training to robustly optimize an objective function. Having said that, I find this to be an interesting paper that utilizes the techniques used in adversarial robustness to design robust solutions for the optimal power flow problem. The paper combines well-known mathematical concepts, such as implicit function theorem and implicit differentiation, in a new way to achieve this goal. The method developed in this work is clearly presented and is backed up with theoretical analysis and experimental evaluation. It might also have applications other than the problem of optimal power flow considered in this paper.

**Time Spent Reviewing:**

6

---

> ### Author Response · Authors · 2021-08-10
> **Response to Reviewer SScg**
>
> Thank you for your interest in our work!

---

### Decision · Program_Chairs · 2021-09-27

**Decision:**

Accept (Poster)

**Comment:**

This paper considers the N-k security-constrained optimal power flow problem using ideas from adversarially robust training. In particular, they design gradient based methods to solve this problem and implement it on N-3 SCOPF which has been traditionally considered out of reach of existing approaches. All the reviewers agree that this paper makes a nice and interesting contribution to this problem and recommend acceptance.